# The gating cycle of a K$^+$ channel at atomic resolution

**Luis G Cuello[1]\*, D Marien Cortes[1], Eduardo Perozo[2]\***

[1]Center for Membrane Protein Research, Department of Cell Physiology and Molecular Biophysics, Texas Tech University Health Sciences Center, Lubbock, United States; [2]Department of Biochemistry and Molecular Biology, The University of Chicago, Chicago, United States

**Abstract** C-type inactivation in potassium channels helps fine-tune long-term channel activity through conformational changes at the selectivity filter. Here, through the use of cross-linked constitutively open constructs, we determined the structures of KcsA's mutants that stabilize the selectivity filter in its conductive (E71A, at 2.25 Å) and deep C-type inactivated (Y82A at 2.4 Å) conformations. These structural snapshots represent KcsA's transient open-conductive (O/O) and the stable open deep C-type inactivated states (O/I), respectively. The present structures provide an unprecedented view of the selectivity filter backbone in its collapsed deep C-type inactivated conformation, highlighting the close interactions with structural waters and the local allosteric interactions that couple activation and inactivation gating. Together with the structures associated with the closed-inactivated state (C/I) and in the well-known closed conductive state (C/O), this work recapitulates, at atomic resolution, the key conformational changes of a potassium channel pore domain as it progresses along its gating cycle.
DOI: https://doi.org/10.7554/eLife.28032.001

\*For correspondence:
luis.cuello@ttuhsc.edu (LGC);
eperozo@uchicago.edu (EP)

**Competing interests:** The authors declare that no competing interests exist.

## Introduction

The simplest description of the gating cycle in the pore domain of a K$^+$ channel requires at least four distinct kinetic states (*Ostmeyer et al., 2013*; *Panyi and Deutsch, 2006*; *Yellen, 1998*). At rest, the pore domain's activation gate formed by the inner helix bundle is closed (C) while the selectivity filter is presumably conductive (O). We define this conformation as the C/O state. When an activating stimulus drives the opening of the channel's inner helix bundle gate, the selective flow of K$^+$ can be sustained for hundredths of milliseconds through the open (O/O) state. However, as a consequence of allosteric coupling between the inner gate and the pore helix near the selectivity filter (*Cuello et al., 2010a*; *Pan et al., 2011*), opening of the activation gate in many channels leads to a series of structural changes at the selectivity filter that renders it non-conductive (the O/I state), a process known as C-type inactivation (*Hoshi et al., 1991*; *Liu et al., 1996*; *López-Barneo et al., 1993*). Once the activating stimulus has ceased, the channel's activation gate returns to the closed conformation and the channel transiently occupies the closed-inactivated (C/I) state with the selectivity filter inactivated. Ultimately, allosteric interactions with the activation gate reset the selectivity filter back to its conductive conformation (C/O), completing the gating cycle (*Figure 1a*).

The high-resolution structures of KcsA in high and low K$^+$ concentrations have been known for a number of years from crystals of KcsA-Fab complexes (*Zhou et al., 2001a*). These structures have been suggested to correspond to the putative C/O (high K$^+$) and C/I (low K$^+$) states (*Zhou et al., 2001b*). Previously, it was determined the structures for KcsA in a putatively partially open-inactivated and in its fully-open and deep-inactivated states at modest resolution by crystallizing a constitutively open mutant (KcsA-OM) displaying different degrees of openings at the activation gate (*Cuello et al., 2010b*). This strategy allowed to propose a set of structural and ion occupancy

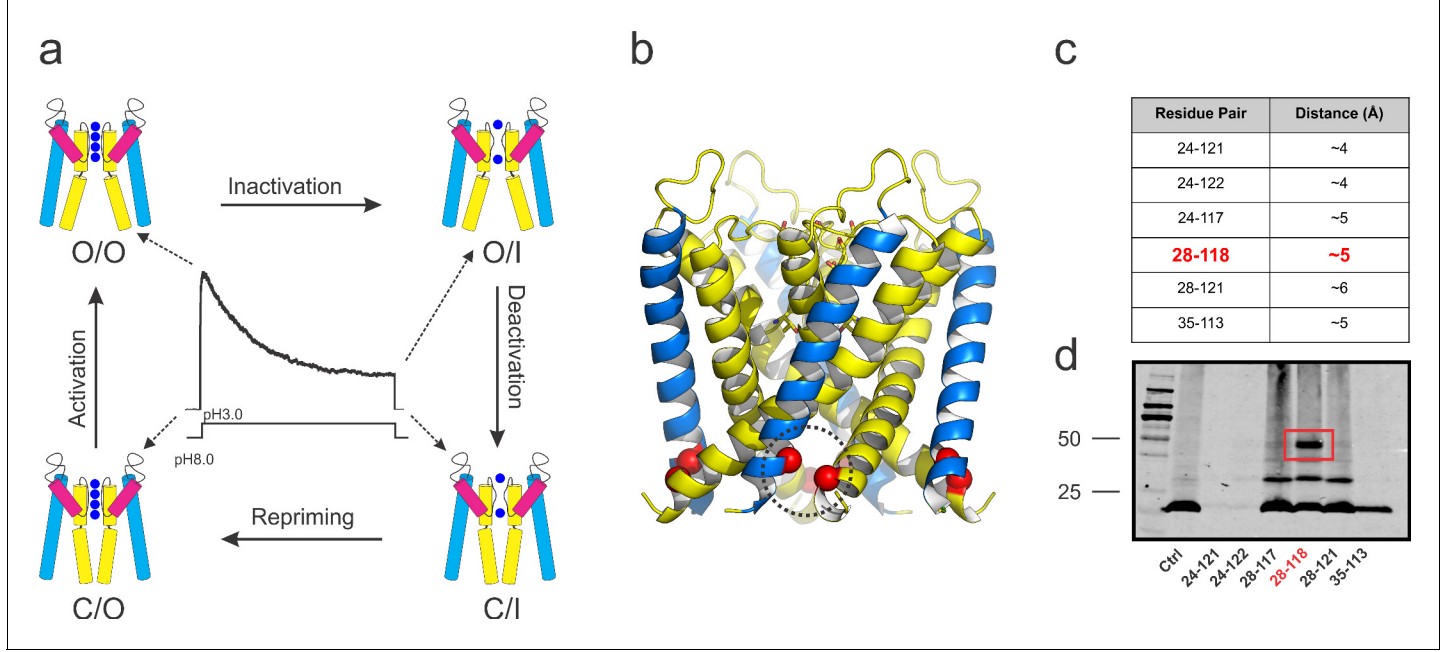

**Figure 1.** KcsA's kinetic cycle and the biochemical strategy to trap of KcsA open gating intermediates. (**a**) pH-dependent macroscopic current recorded from asolectin liposomes containing KcsA *wild type*, surrounded by cartoon representations of the structural equivalents for the gating intermediates. KcsA's gating cycle comprises at least four distinct kinetic states defined by gating intermediates in which the selectivity filter is conductive (O) or collapsed (I) and the activation gate is open (O) or closed (C). Transitions between these kinetic states define four different gating processes: C/O→O/O, that is, activation; O/O→O/I, that is, C-type inactivation; O/I→C/I, that is, deactivation and C/I→C/O, that is, repriming or recovery from C-type inactivation. (**b**) side views of KcsA's O/I low-resolution structure (PDB, 3F5W) displaying the region of the channel that was tested for inter-subunit disulfide-bond formation (red spheres). (**c**) table indicating all the putative residue-pairs tested for cysteine-bridge formation. In red, the pair 28–118 spontaneously formed disulfide-bonds. (**d**) A western blot showing that the residue pair 28–118 formed a covalently linked tetramer upon introduction of cysteine residues at those positions (the samples were heated in the absence of a reducing agent, non-reducing conditions).
DOI: https://doi.org/10.7554/eLife.28032.002

changes in the selectivity filter associated with KcsA inactivation gating. The observed changes involve an initial constriction of the permeation pathway at G77, between $K^+$ binding sites S2 and S3 at the selectivity filter. This change is accompanied by a concomitant reduction in ion occupancy in S2 and it was proposed represents an early non-conductive state ($I^1$). As the inner helical bundle gate opens further, there is additional narrowing of the selectivity filter at G77, causing the collapse of the permeation pathway and the loss of $K^+$ at both, S2 and S3 binding sites, which likely represents a deep C-type inactivated state of KcsA or $I^2$ (*Cuello et al., 2010b*). The existence of a deep C-type inactivated conformation of KcsA's selectivity filter ($I^2$ state) is supported by solution and solid state Nuclear Magnetic Resonance spectroscopy experiments (*Ader et al., 2009*; *Ader et al., 2008*; *Bhate et al., 2010*; *Imai et al., 2010*; *Wylie et al., 2014*) and 2DIR measurements (*Kratochvil et al., 2016*). However, at ~3.2 Å resolution, the original O/I (*Cuello et al., 2010b*) structure does not provide information on the position of the carbonyl groups responsible for ion coordination (*Zhou et al., 2001b*).

Further analyses of these structures revealed that most of the degradation in overall resolution was likely due to conformational heterogeneity at the activation gate formed by the inner helix bundle (*Cuello et al., 2010b*). In an attempt to increase the resolution of pore domain states where the inner helix bundle gate is open, a set of cysteine-bridges between adjacent subunits was engineered, using the lower resolution structure of KcsA-OM as a template (*Cuello et al., 2010b*). This experimental approach, combined with targeted mutations known to affect the stability of the C-type inactivated conformation (*Cordero-Morales et al., 2006*), allowed the structural determination of KcsA in its 'transient' open-conductive (O/O, mutant E71A) and deep C-type inactivated (O/I, mutant Y82A) conformations at near atomic resolutions (2.25 Å and 2.4 Å, respectively).

## Results

### Trapping KcsA with its activation gate in the open conformation

Using the open-inactivated structure of KcsA (PDB = 3F5W) (*Cuello et al., 2010b*) as a reference (*Figure 1b*), six cysteine pairs were tested between the N-terminal side of the transmembrane segment 1 (TM1) and the C-terminal end of TM2 (*Figure 1c*) for disulfide-bond formation in SDS gels (*Figure 1d*). While a number of cysteine pairs were formed under oxidizing conditions, cysteine-pair 28–118 (between adjacent subunits) spontaneously formed a covalently linked tetramer, which presumably has trapped-open KcsA's activation gate (Locked-open KcsA). This double-cysteine mutant, while displaying similar overall biochemical properties to those of wild type KcsA (*Figure 2a*) is also resistant to high temperatures, as indicated by thermal denaturation experiments (*Figure 2b*). Single-channel activity recorded on liposomes containing Locked-open KcsA (in symmetrical 200mM-K$^+$) exhibited low steady-state open probability and large single-channel conductance (~200 pS) (*Figure 2c*), as expected from channels mostly populating the O/I state. With Rb$^+$ as charge carrier, open probability increases and the single-channel conductance is reduced (~23 pS) (*Figure 2d*). These changes are the result of ion-specific interactions at the selectivity filter (*Lockless et al., 2007*; *Morais-Cabral et al., 2001*; *Zhou et al., 2001a*) and suggest that indeed, most of the gating transitions obtained in these steady-state single-channel recordings derive from conformational transitions at the selectivity filter.

### Locked-open KcsA and mutations that modulate C-type inactivation

It was shown earlier that mutations nearby KcsA-selectivity filter can either eliminate (E71A) or strengthen (Y82A) C-type inactivation gating (*Figure 3a, b and c*). Hence, these mutants were used to bias the Locked-open KcsA gating behavior by incorporating mutations proven to modulate C-type inactivation gating in KcsA. As predicted, the E71A mutant on the Locked-open channel background displayed a pH-independent high-open probability single- channel activity (*Figure 3d*

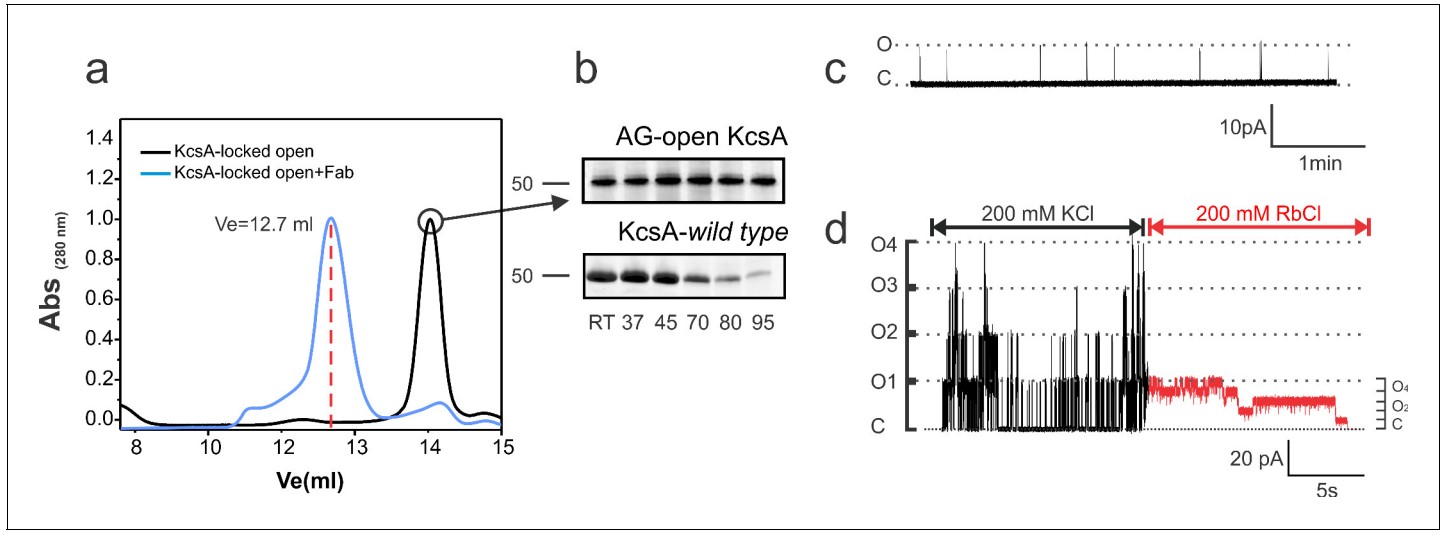

**Figure 2.** Biochemical and functional characterization of the Locked-open KcsA. (a) Hydrodynamic properties of the disulfide-bonded KcsA-OM (Locked-open KcsA) were assessed by analytical size exclusion chromatography in a Superdex HR200 column. The chromatogram shows, in black, the elution volume for the chymotrypsin truncated Locked-open KcsA and in blue, in complex with a Fab fragment (Ve = 12.7 ml) required for crystallization trials. (b) Since the Locked-open-KcsA is covalently linked by disulfide bonds, its tetrameric state is remarkably resistant to high temperature (upper panel), when compared to KcsA wild type (lower panel). Melting experiments were conducted for 30 min at the temperature indicated in the absence of a reducing agent. (c) A representative single-channel trace of the Locked-open KcsA channel in 200 mM KCl, displays a very low-open probability, which agrees with it being an open and C-type inactivated channel. (d) An excised multi-channel patch of the Locked-open KcsA displays the characteristic low-open probability mode and large single- channel conductance when K$^+$ was the permeant ion, 200 mM KCl (Black). After rapidly switching the bath solution to Rb$^+$ as the permeant ion, 200 mM RbCl (Red), the channel switches to a high-open probability mode and small single-channel conductance.

DOI: https://doi.org/10.7554/eLife.28032.003

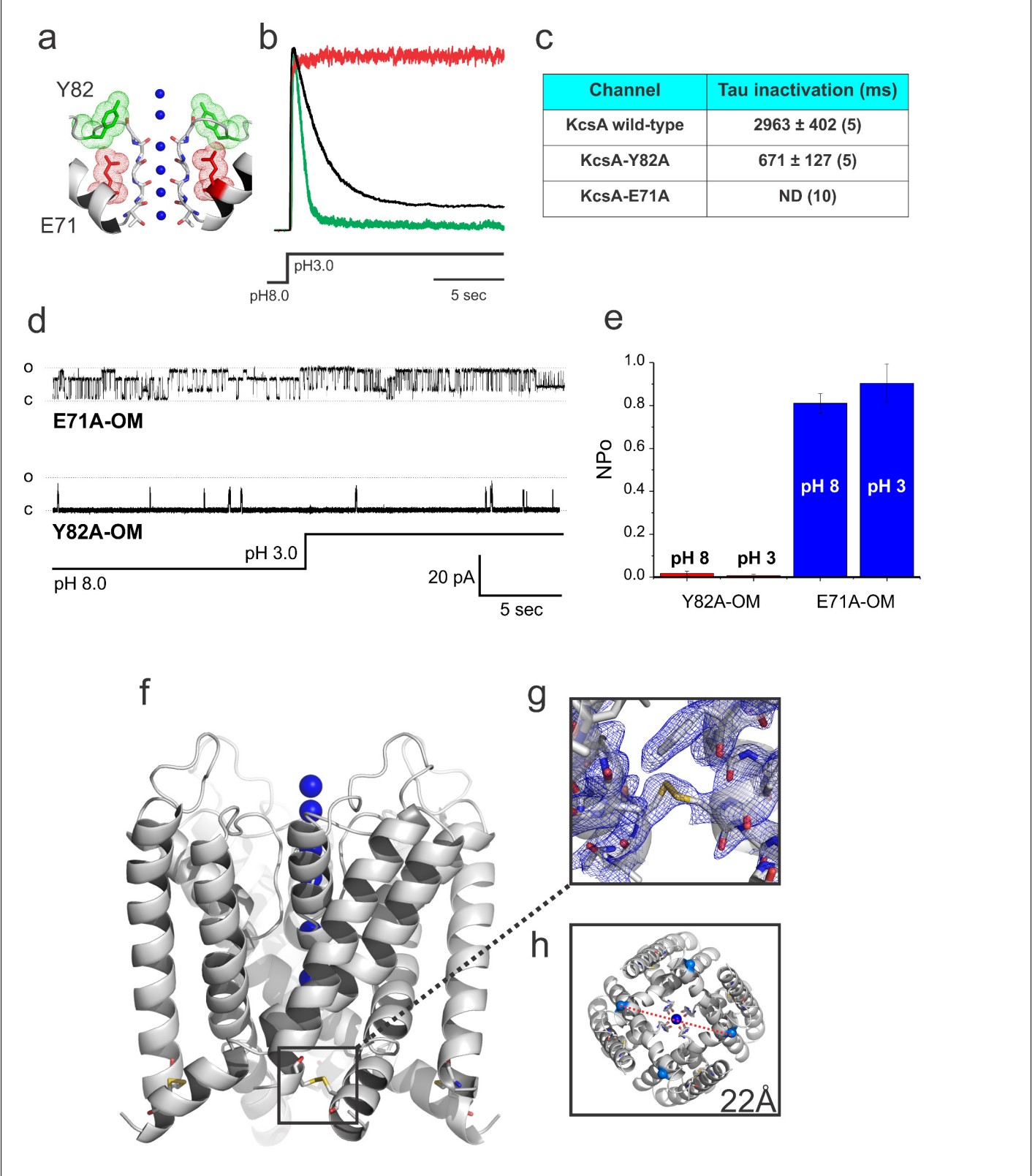

**Figure 3.** Crystal structure of the Locked-open KcsA. (a) A cartoon representation of KcsA's selectivity filter indicating mutated positions used in this study that either accelerates (Y82A in green) or removes (E71A in red) C-type inactivation. (b) Representative macroscopic currents recorded in symmetric 200 mM KCl during a pH fast-switching experiment for: the wild type channel (black), the fast-inactivating mutant Y82A (green) and the non-inactivating E71A mutant (red). (c) Tau for inactivation of KcsA wild type, Y82A and E71A mutants after a pH jump experiment (reported values are an

*Figure 3 continued on next page*

*Figure 3 continued*

average of the number (n) of independent observations indicated in parenthesis). (**d**) Representative 30 s. single-channel recordings of the Locked-open E71A (upper trace) and Y82A (lower trace) mutants, the pH of the bath solution was switched from 8 to 3 after 15 s. of recording by using a fast perfusion system. (**e**) The number of channel x open probability (NPo) of the Locked-open channel, as predicted, was pH insensitive but high for the E71A (~0.8–0.9) and low for the Y82A mutants (~0.015–0.04) Reported values are means ±SEM of 3 independent experiments. (**f**) A cartoon representation of the Locked-open KcsA X-ray structure highlighting the location of the inter-subunit disulfide bond between introduced cysteine residues at position 28 and 118 (**g**) 2Fo-Fc electron-density map (cyan, contoured at 1.0σ) of the Locked-open KcsA validating the formation of the 28cys-118cys disulfide bond. (**h**) An intracellular view displaying an activation gate opening of 22 Å (measured between Threonine 112 α-carbons on diagonal subunits).

DOI: https://doi.org/10.7554/eLife.28032.004

*and e*), albeit showing discernible subconductance states (attributed to an incomplete cross-linking of the activation gate). Nevertheless, the E71A Locked-open seems to effectively sample the O/O state. Conversely, the Y82A mutant on the Locked-open channel background exhibited a pH-independent low-open probability single- channel activity (*Figure 3d and e*) and presumably is sampling the O/I state. These results provided an experimental path towards the crystallization of KcsA's O/O and O/I states at improved resolutions.

## Atomic-resolution structure of KcsA open and conductive

Incorporating the non-inactivating mutation E71A on the Locked-open-KcsA scaffold would lead to an efficient sampling of the elusive O/O state (*Cordero-Morales et al., 2006*) (*Figure 3a, d and e*). Indeed, crystals of the Locked-open-KcsA E71A mutant diffracted to 2.25 Å Bragg spacings, and the structure was solved by molecular replacement with $R_{work}$ = 0.186 and $R_{free}$ = 0.213, PDB = 5VK6 (*Table 1*). The O/O state electron-density map validated the inter-subunit disulfide bond between positions 28 and 118 (*Figure 3f and g*). In the O/O state the activation gate spans 22 Å in diameter (Cα-Cα distance at Thr 112) (*Figure 3h*), in agreement with the more physiologically relevant open conformation found in the full-length structures of KcsA and Kv1.2 channels (*Uysal et al., 2011*; *Long et al., 2005*).

As expected, in the O/O state the selectivity filter shows 4 $K^+$ ions in the canonical 1,3 and 2,4 configurations (*Figure 4a*), together with three more $K^+$ ions, two at the extracellular entrance and one inside the channel's central cavity (*Figure 4b*). This is in agreement with most predictions suggesting that the O/O and C/O-state conformations should be similar around the selectivity filter. At the present resolution, there were identified four water molecules coordinating the $K^+$ ion in the central cavity, consistent with the coordination of a partially dehydrated ion about to enter the selectivity filter (*Figure 4b and c*). Superimposing the selectivity filter structures of KcsA C/O and O/O states (*Figure 4d*) confirms the conformational equivalence of these structures within 0.26 Å RMSD (for selectivity filter atoms). Furthermore, although the activation gate is open, the overall organization of the eight oxygen atoms surrounding each $K^+$ ion is similar in the C/O and O/O states.

Despite these similarities, some interesting differences were found between the C/O and O/O states, when we compare the B-factors of the selectivity filter in the C/O and O/O states (*Figure 5*). However, to make a relevant B-factor comparison between different structures they need to share similar resolutions (*Balendiran et al., 2014*). To that end, a new structure for KcsA's closed state was solved by molecular replacement at the same nominal resolution (2.25 Å) of the Locked-open E71A mutant using similar parameters during the refinement process. The normalized B-factors of the pore helix residues (Cα) in the O/O and the C/O states were very similar and displayed similar periodic behavior but largely digress from each other within the selectivity filter residues (*Figure 5*). These experimental observations strongly suggest that the strength of the interaction of the carbonyl groups from the peptide backbone and the $K^+$ ions within KcsA's selectivity filter could be different for the C/O and the O/O states.

Interestingly, significant changes in ion selectivity during activation gating have been measured in voltage-gated $K^+$ channels (*Zheng and Sigworth, 1997*). Additionally, it has been suggested that the selectivity filter structures of the closed and open states in $K^+$ channels should differ slightly depending on the conformation in the filter as the inner gate opens (*Chapman and VanDongen, 2005*; *Chapman et al., 1997*; *Zheng and Sigworth, 1997*; *Zheng and Sigworth, 1998*).

**Table 1.** Crystallographic table and refinement statistics.

| Statistic | E71A-OM | Y82A-OM | Y82A-F103A closed |
|---|---|---|---|
| Data Collection | | | |
| Space Group | I4 | I4 | I4 |
| Cell Dimension | | | |
| $a = b, c$ (Å) | 156.54, 74.61 | 156.19, 74.17 | 155.73, 76.225 |
| $\alpha=\beta=\gamma$ (°) | 90 | 90 | 90 |
| Resolution (Å) | 24.75–2.25 (2.33–2.25) | 36.82–2.37 (2.45–2.37) | 42.91–2.25 (2.33–2.25) |
| $R_{merge}$ | 0.052 (0.38) | 0.059 (0.46) | 0.079 (0.26) |
| I/σI | 29.3 (3.9) | 29.53 (2.9) | 15.8 (2.9) |
| Completeness (%) | 95.11 (99.70) | 99.76 (99.24) | 98.43 (91.97) |
| Redundancy | 4.6 (5.0) | 4.5 (4.4) | 4.5 (4.4) |
| Refinement | | | |
| No. reflections | 42598 (4255) | 36377 (3605) | 42773 (3985) |
| $R_{work}$ | 0.186 (0.228) | 0.194 (0.239) | 0.174 (0.219) |
| $R_{free}$ | 0.213 (0.262) | 0.226 (0.265) | 0.203 (0.234) |
| No. atoms | 4088 | 4054 | 4386 |
| Protein | 3939 | 3926 | 4036 |
| Ligand/ion | 7 | 3 | 6 |
| Waters | 101 | 84 | 218 |
| Other ligands | 48 | 44 | 47 |
| Protein residues | 529 | 528 | 535 |
| Bond lengths (Å) | 0.003 | 0.003 | 0.01 |
| Bond angles (°) | 0.64 | 0.65 | 1.02 |
| Wilson B-factor | 50.05 | 51.10 | 30.63 |
| Average B-Factor, Å$^2$ | 69.09 | 67.94 | 39.39 |
| Protein | 69.22 | 68.08 | 38.93 |
| Ligands | 79.42 | 76.41 | 49.72 |
| Water | 59.26 | 75.05 | 45.37 |
| Ramachandran Favored (%) | 97.35 | 95.97 | 97 |
| Ramachandran outliers (%) | 0.19 | 0.38 | 0.19 |

*Highest resolution shell is show in parenthesis.

[†]Data sets were collected from a single crystal.

DOI: https://doi.org/10.7554/eLife.28032.005

Clearly, the structure of the Locked-open KcsA-E71A mutant represents a snapshot of the pore domain of a K$^+$ channel only transiently available during normal gating cycle operation. KcsA's O/O state is therefore likely to adopt a filter conformation similar to that found in the present Locked-open KcsA E71A structure. However, while this experimental manipulation led to a structure with an activation gate open and a conductive selectivity filter, this is only a condition biased by disulfide bonds and is unlikely to reflect potential conformational dynamic changes as a consequence of ion fluxes and/or the presence of a membrane potential.

## Atomic-resolution structure of KcsA open and deep C-type inactivated

In most K$^+$ channels the O/O conformation is metastable, as it eventually transitions into the O/I or C-type inactivated state. To further bias the selectivity filter into this conformation it was engineered onto the Locked-open KcsA scaffold a mutant which exhibits an increased rate and steady-state level of C-type inactivation(*Cordero-Morales et al., 2006*; *López-Barneo et al., 1993*). In KcsA, mutant Y82A (equivalent mutation to T449A in *Shaker*) also leads to an augmented rate and degree of

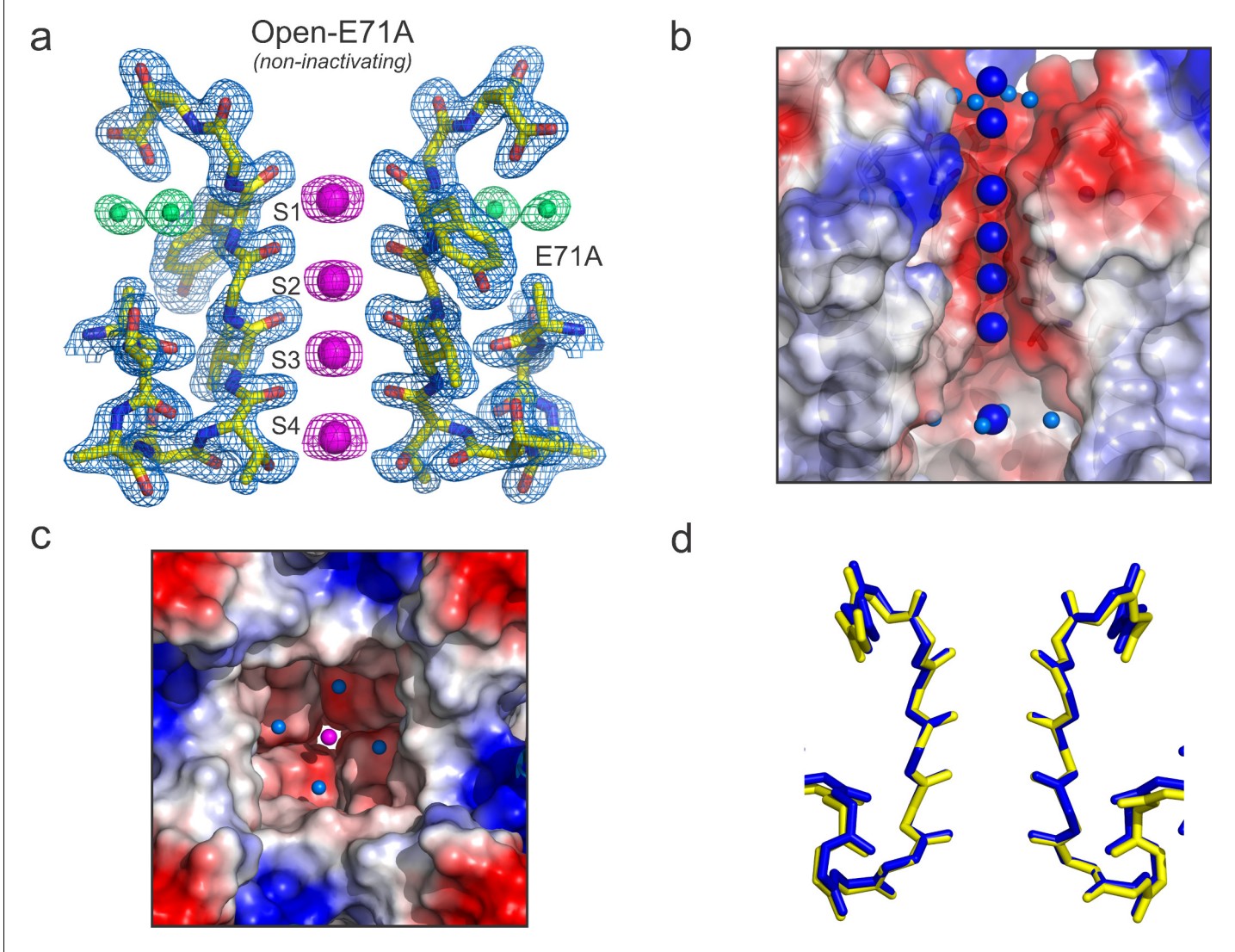

**Figure 4.** High-resolution structure of the open and conductive state of KcsA. (a) X-ray structure of the Locked-open KcsA E71A mutant, determined @ 2.25 Å resolution. Displayed is the 2Fo-Fc electron-density map (light-blue mesh contoured at 2.5σ) for the selectivity filter residues 70–80 from two diagonally symmetric subunits, colored in yellow with oxygen atoms in red; for the $K^+$ ions within the filter (magenta mesh contoured at 2.6σ) and for two water molecules located behind the channel's selectivity filter that are occupying the vacancy created by the glutamate to alanine substitution (green mesh contoured @ 3σ). (b) Side view of a surface representation of KcsA O/O state (Locked-open-E71A mutant) highlighting the $K^+$ ions within the channel pore. At the extracellular side two $K^+$ ions are observed, one of them (the most internal) is partially dehydrated and coordinated by four water molecules. Similar to the closed KcsA structure (1K4C), the O/O state also contains a $K^+$ ion within the channel central cavity but in contrast to the closed state in which is coordinated by eight water molecules, the ion is partially dehydrated and tetra-coordinated. (c) Intracellular view of the KcsA O/O state showing a $K^+$ ion, inside the channel central cavity, partially dehydrated and coordinated by four water molecules (d) Structural alignment of the selectivity filters of the Locked-open KcsA E71A mutant (PDB 5VK6) in blue and the closed-KcsA in yellow (PDB 1K4C). The atomic deviation between the peptide backbones of the two filters (r.m.s.d.) is 0.19 Å.

DOI: https://doi.org/10.7554/eLife.28032.006

C-type inactivation (*Cordero-Morales et al., 2006*) (*Figure 3a*, green trace), which it could be interpreted as favoring entry into a deep C-type inactivated state (a collapsed selectivity filter). Crystals of the O/I-state diffracted ~2.4 Å and its structure was solved by molecular replacement ($R_{work}$ = 0.194 and $R_{free}$ = 0.226) PDB = 5 VKE (*Figure 6a*). This represents the highest resolution structure of KcsA's O/I state available, offering a unique perspective on the interactions between the selectivity filter backbone, permeant ions, and structural waters that stabilize its deep collapsed state.

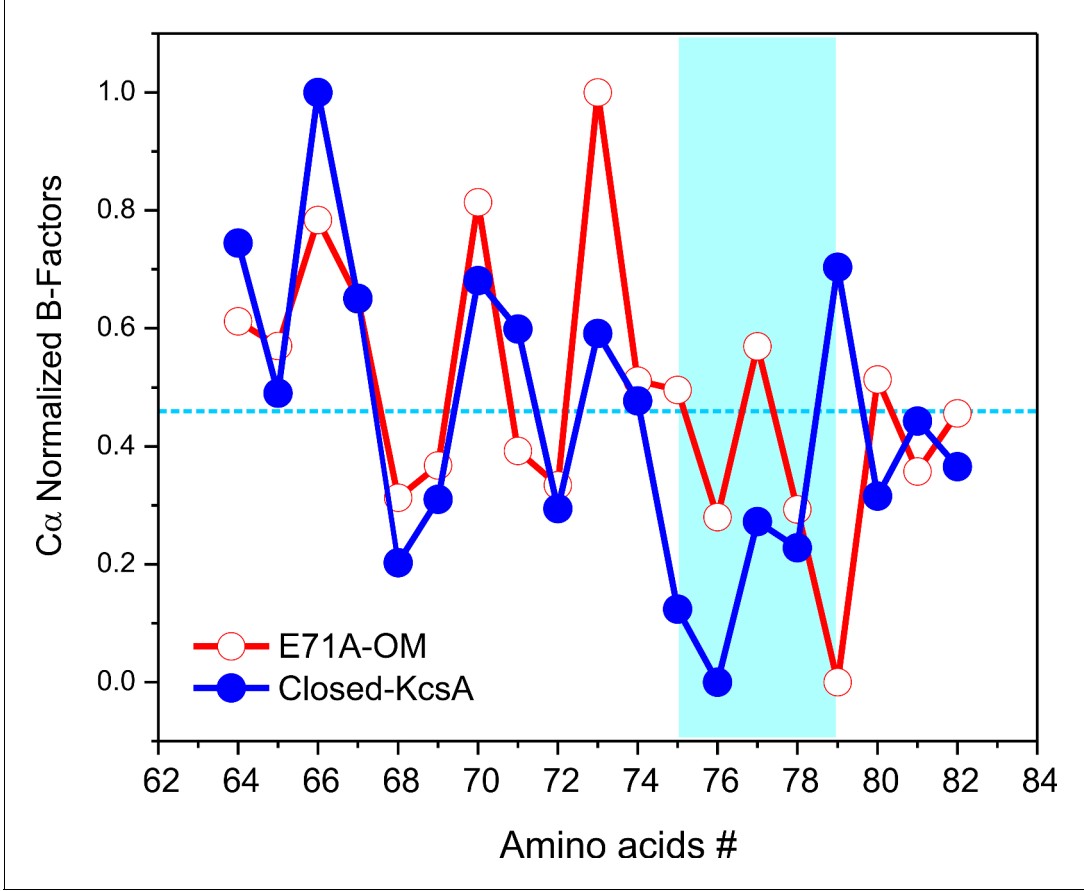

**Figure 5.** Crystallographic normalized B-factors for KcsA pore helix and selectivity filter alpha-carbons (Cα). Changes of the normalized crystallographic B-factors of KcsA's pore helix and selectivity filter residues (*signature sequence*) during activation gating are best represented by: (a) KcsA closed-primed-to-conduct or C/O state structure (new structure solved at the same resolution and using the same parameters during refining than for the O/O state structure) and KcsA open-conductive or O/O state structure (PDB 5VK6).

DOI: https://doi.org/10.7554/eLife.28032.007

In the O/I-state, the selectivity filter contains 2 $K^+$ ions tightly coordinated at positions 1 and 4, with ion vacancies at positions 2 and 3 (*Figure 6a*). The backbone carbonyl of Val-76 flips out of the axis of symmetry, obliterating the S2 and S3 $K^+$ binding sites, as this transition directly affects $K^+$ coordination by backbone carbonyls at the center of selectivity filter. This seems to represent an insurmountable energetic barrier to ion movement along KcsA-selectivity filter (*Cuello et al., 2010b*). Interestingly, Val-76 carbonyls can be seen forming hydrogen bonds with newly resolved structural waters behind KcsA's selectivity filter (*Figure 6a*). These appear to be a feature of the collapsed filter conformation and should be considered when analyzing the energetics of C-type inactivation onset and repriming. Moreover, as originally reported at lower resolution (and also observed in the low [$K^+$] KcsA structure), the O/I-state pore is collapsed at the Gly-77 (*Figure 6a*), where the inter-subunit Cα diameter is reduced from 8 to 6 Å. This change originates from the rotational freedom of Gly-77 α-carbons, and represents an additional steric barrier to the movement of $K^+$ ions along the selectivity filter (*Cuello et al., 2010a*). Contrary to what is seen in the C/O or O/O states, the O/I-state displays a minor change in the pitch of the last turn of the pore helix (~0.5 Å shorter, [*Figure 6—figure supplement 1*]) (*Cuello et al., 2010a*), creating an expansion of the S3 and S4 $K^+$ binding sites. This selectivity filter is structurally reminiscent to selectivity filters in both, a constitutively open-inactivated mutant (*Cuello et al., 2010a*) or in the KcsA closed structure at low [$K^+$] concentration (*Ader et al., 2008*; *Bhate and McDermott, 2012*; *Wylie et al., 2014*; *Zhou et al., 2001b*).

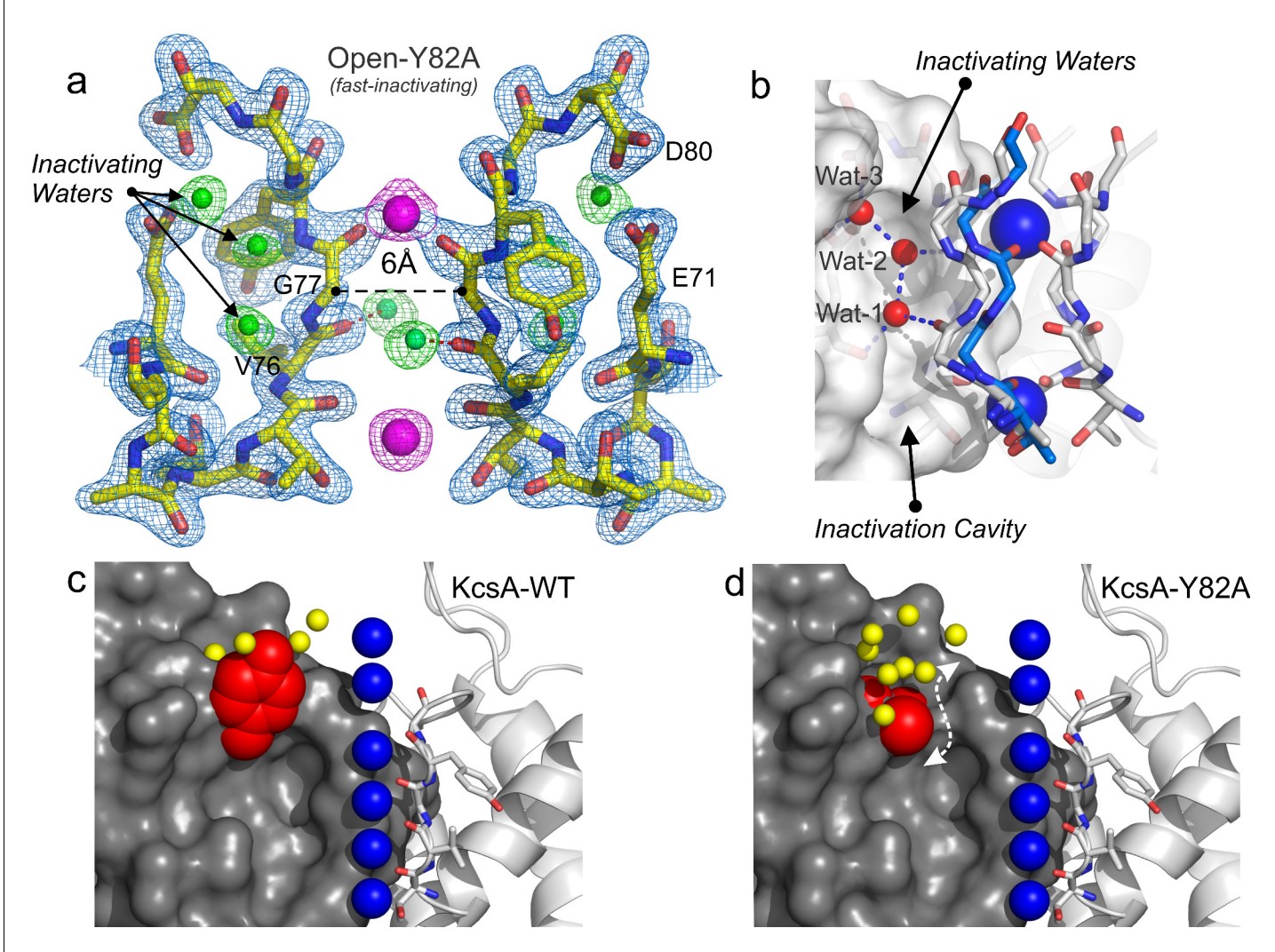

**Figure 6.** Structure of KcsA open and deep C-type inactivated. (a) X-ray crystal structure of the Locked-open open and fast inactivating Y82A mutant @ 2.37 Å resolution (PDB 5VKE). Shown is the 2Fo-Fc electron-density map for the selectivity filter residues 71–80 from two diagonally symmetric subunits, colored in yellow with oxygen atoms in red (light-blue mesh contoured at 2.6σ); for the two remaining ions in the filter (magenta mesh contoured at 3σ) and for three water molecules, 'inactivating waters', located inside a cavity behind KcsA filter (green mesh contoured at 3σ). The Cα-Cα distance at position Gly 77 measured from diagonal subunits is 6 Å, narrowing KcsA's filter ~2 Å. (b) Selectivity filters of the closed-KcsA (PDB 1K4C) colored in blue and oxygen atoms in red as a reference and of the Locked-open KcsA fast-inactivating Y82A mutant (PDB 5VKE) colored in white and oxygen atoms in red. The peptide backbone carbonyl groups from the Locked-open KcsA Y82A mutant flips away from the channel's axis of symmetry moving sideways into a cavity, (b) 'inactivation cavity', which is located behind the channel's filter of a neighboring subunit. This network of water molecules holds the filter in the inactive or flipped conformation. (c) Cartoon representation of KcsA wild type (PDB 1K4C), colored in white with oxygen atoms in red, as a reference and on the left a surface representation of a neighboring subunit in which the entryway of the inactivation cavity is guarded by Tyr 82 (in red), which is positioned to putatively regulate the rate and the extent of C-type inactivation by steric regulation and or/hydrophobic effect of the diffusion of water molecules (yellow spheres) into the 'inactivation cavity'. (d) X-ray crystal structure of the Y82A mutant in the closed state @ 2.25 Å resolution. Substitution of the bulky and hydrophobic tyrosine at position 82 by alanine not only eliminates the steric limitation of water flow into the inactivation cavity (dashed arrow) but it solvates the entryway of the cavity, as denoted by the large number of crystallographic water molecules identified at this region of the structure (yellow spheres).

DOI: https://doi.org/10.7554/eLife.28032.008

The following figure supplement is available for figure 6:

**Figure supplement 1.** KcsA O/I state and the compression of its pore helix last turn.

DOI: https://doi.org/10.7554/eLife.28032.009

At the higher resolution of the Locked-open-Y82A structure, a novel network of water molecules appears behind KcsA's selectivity filter. These three 'inactivating waters' were predicted on computational grounds (*Ostmeyer et al., 2013*) and appear to stabilize the O/I conformation (*Figure 6a and b*). Val-76 C = O has moved away from the axis of symmetry and is hydrogen-bonded to one of the inactivating water molecules (wat-1), which resides in an 'inactivation cavity' behind KcsA's selectivity filter (*Figure 6b*). Wat-1 is in turn hydrogen-bonded to wat-2 and wat-3 and to other chemical groups of the selectivity filter, from bottom to top: Glu-71 carbonyl, Val-76 carbonyl in a neighboring subunit, Gly-77 amide nitrogen, Glu-71 and Asp-80 side chains (*Figure 6b*).

## Diffusion of 'inactivating waters' determines the rate of C-type inactivation gating

The existence of these inactivating water in KcsA's deep C-type inactivated state is highlighted by both, the arrangement of hydrogen-bonds formed with the backbone carbonyls in the selectivity filter and their absence in the conductive state of the filter (*Zhou et al., 2001b*). Consequently, water access to the inactivation cavity becomes a critical factor controlling the rate and magnitude of this process. In this mechanism, water molecules would diffuse into the inactivation cavity by a conduit formed by the apposition of two neighboring subunits. The entrance to this conduit (*Figure 6c*) is guarded by Tyr-82 (Thr-449 in Shaker). This position, at the external mouth of the filter has been shown to regulate C-type inactivation gating in a variety of $K^+$ channels (*Cordero-Morales et al., 2006*; *López-Barneo et al., 1993*). For instance, replacing Tyr 82 in KcsA with Alanine accelerates C-type inactivation gating (*Cordero-Morales et al., 2006*; *López-Barneo et al., 1993*), while in *Shaker* T449Y slows down this process. These experimental observations suggest that this position could work as a gatekeeper, regulating the diffusion of water molecules into the inactivation cavity. Thus, a reduction in the volume and/or hydrophobicity of this amino acid side chain would maximize the diffusion of water molecules into the inactivation cavity, increasing the rate and the magnitude of C-type inactivation. In order to address this possibility, the crystal structure of the closed Y82A mutant was solved at 2.25 Å resolution, PDB = 5 VKH, ($R_{work}$ = 0.1749 and $R_{free}$ = 0.2037). As expected, a smaller and less hydrophobic amino acid at this position not only creates a septum or a channel that communicates the extracellular milieu and the 'inactivation cavity', possibly enhancing the diffusion of water molecules into it, but also allows the solvation of the inactivation cavity entryway, as evidenced by the additional crystallographic water molecules detected at this region (*Figure 6d*). Interestingly, it has been shown computationally and experimentally that recovery from C-type inactivation is in fact, dependent on the occupancy of this buried waters into the 'inactivation cavity' and that as predicted, changing the water activity at the extracellular part of the channel slows down KcsA's inactivation kinetics (*Ostmeyer et al., 2013*).

Finally, the central cavity of the O/I state holds a density, putatively modeled as a $K^+$ ion, since the crystallization solution contains at least 300 mM KCl. However, even at the present resolution no discernible waters were observed coordinating the ion inside the cavity, as it was the case in the O/O state. However, a large number of crystallographic water molecules were found likely associated with the changes in surface polarity or H-bond availability of the central cavity in the C-type inactivated conformation (*Figure 7*).

## Discussion

In KcsA, once the activation gate opens, C-type inactivation is initiated by the loss of $K^+$ coordination at the S2 binding site due to the narrowing of the permeation pathway at Gly 77 (*Cuello et al., 2010b*) (*Figure 6a*). Further constricting the selectivity filter at Gly 77 causes structural changes that eventually leads to a collapsed, deep C-type inactivated conformation of the filter with the loss of S2 and S3 $K^+$ coordination sites. This mechanism of C-type inactivation has found support from a variety of spectroscopic and computational studies carried out in KcsA (*Ader et al., 2009*; *Ader et al., 2008*; *Bhate and McDermott, 2012*; *Bhate et al., 2010*; *Imai et al., 2010*; *Ostmeyer et al., 2013*; *Wylie et al., 2014*; *Kratochvil et al., 2016*). However, recent studies based on crystal structures of refolded semisynthetic closed-KcsA containing unnatural amino acids, have suggested that the collapsed structure of the selectivity filter might not represent the deep C-type-inactivated conformation in KcsA (*Devaraneni et al., 2013*).

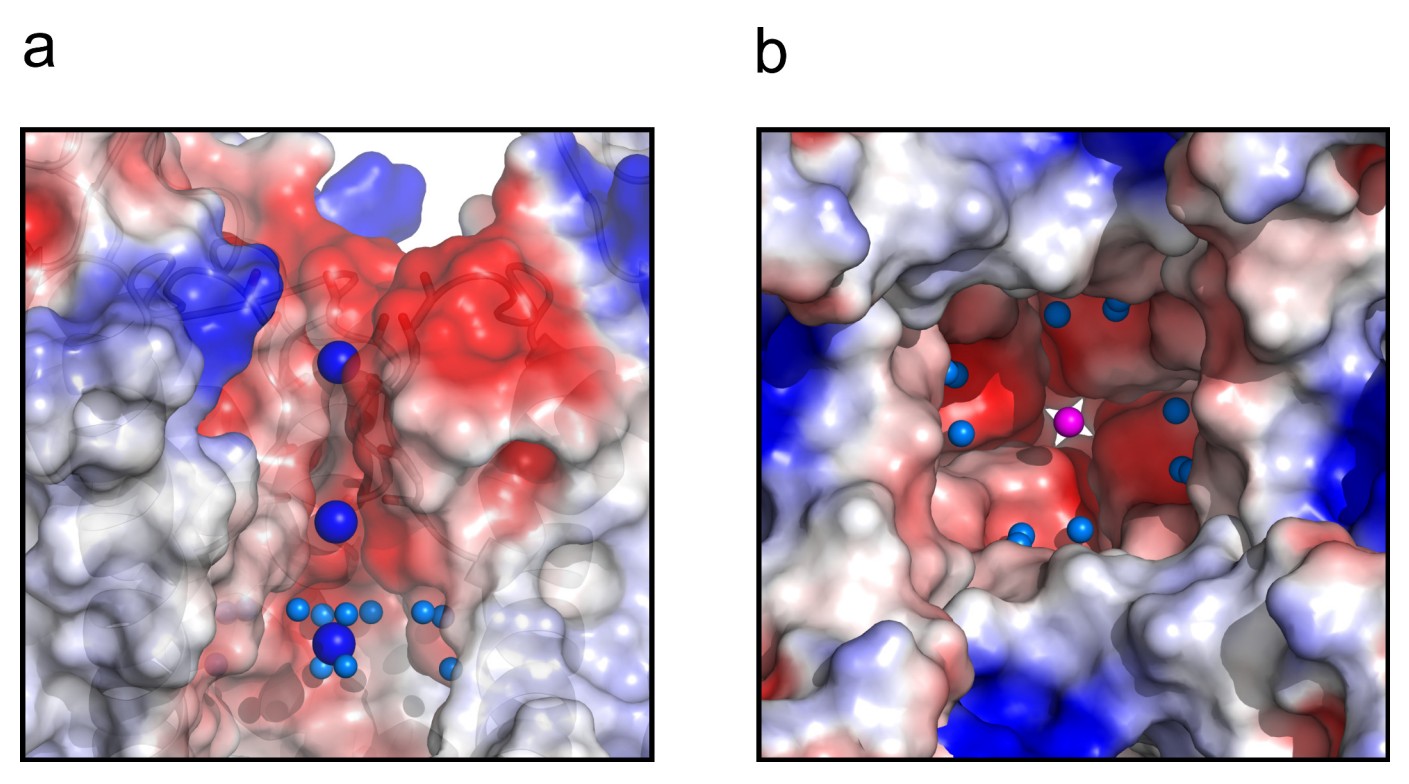

**Figure 7.** The central cavity of the KcsA O/I state. (**a**) A side view of a surface representation of KcsA O/I state or Locked-open KcsA Y82A mutant crystal structure. KcsA O/I state seems to hold a K$^+$ ion (blue sphere) with no discernible coordinating water molecules surrounding it. (**b**) An intracellular view of the central cavity displays a large number of crystallographic water molecules (cyan spheres) by comparison to the cavity of the O/O state or Locked-open KcsA E71A mutant.

DOI: https://doi.org/10.7554/eLife.28032.010

The atomic-resolution crystal structures of KcsA's O/O and O/I states strongly suggest otherwise, since the well-established non-inactivating E71A mutant (*Cordero-Morales et al., 2006*) prevented the collapse of the selectivity filter. This finding indicates that KcsA in the deep C-type inactivated state has a collapsed selectivity filter, as previously suggested (*Cuello et al., 2010a*). Additionally, the existence of time dependent-conductance changes in a D-Ala semisynthetic construct cannot be used as evidence against the structural correlation between the collapsed conformation of the filter and the process of KcsA's inactivation. Not only do these studies vastly underestimate the intrinsic conformational dynamics of the selectivity filter, but recent 2DIR data points towards a more complex conformational landscape for the selectivity filter that might include additional non-conductive conformations (*Kratochvil et al., 2016*). Indeed, recent computational analyses suggest that the D-Alanine 'rigidized' selectivity filter could relax to a partially constricted conformation, with the D-Ala side chains at position 77 occluding the permeation pathway (*Li et al., 2017*). Therefore, it is possible that the 77D-Ala selectivity filter displayed a time-dependent lack of activity due to alternative constricted conformations of KcsA's filter.

Furthermore, these experimental results, together with a number of recent reports in KcsA and other channels (*Ader et al., 2009*; *Ben-Abu et al., 2009*; *Imai et al., 2010*; *Panyi and Deutsch, 2006*; *Peters et al., 2013*) have demonstrated that the key driving force behind KcsA's inactivation is an allosteric coupling between their activation gate and the selectivity filter. This strongly suggests that to properly study the structural changes associated with C-type inactivation at the selectivity filter of K$^+$ channels, a channel must be trapped with its activation gate open, since the structural integrity of a conductive selectivity filter is influenced by a number of parameters, most importantly, the conformation of the activation gate due to their structural coupling. This coupling ultimately will strongly influence the energetics of interaction between permeant ions and the channel's selectivity

filter as well as the pKa of activation and deactivation gating. These observations underlie a novel hysteretic gating behavior in KcsA (*Tilegenova et al., 2017*). Additionally, the non-inactivating Locked-open-KcsA-E71A mutant structure is consistent with the collapsed filter hypothesis of C-type inactivation, since the open state displayed a non-collapsed selectivity filter in this inactivation-deficient mutant.

Recapitulating existing data with the present high-resolution structures, KcsA's C-type inactivation can be explained by the following minimal mechanistic model (*Figure 8—figure supplement 1*). Upon activation gating (C/O-to-O/O transition) KcsA's inner bundle gate opens, expands its diameter to ~22 Å and allowing the diffusion of hydrated $K^+$ ions into the channel central cavity. Once in the cavity, a $K^+$ ion sheds off water molecules (*Figure 4b*) immediately before entering a selectivity filter that has undergone subtle but distinctive structural changes. The O/O is a transient kinetic state that conducts $K^+$ ions for hundredths of milliseconds. But a strong allosteric coupling between KcsA's and the selectivity filter (Phe 103 clashes with I100, Thr 74 and 75 from the signature sequence) deforms the pore helix and as a consequence the selectivity filter undergoes compensatory structural changes (*Pan et al., 2011*; *Cuello et al., 2010a*). These changes involve (1) flipping of the selectivity filter backbone C = O, which triggers the consecutive loss of the second and third $K^+$ binding sites as a function of the channel's activation gate opening and (2) in the absence of the second and third $K^+$ binding sites the selectivity filter collapses. A KcsA kinetic intermediate that has loss as much as one $K^+$ binding site at the selectivity filter is considered an inactivated state ($I_1$) (*Figure 8*, *Figure 8—figure supplement 1*), which can dramatically reduce the unitary conductance of the channel(*Morais-Cabral et al., 2001*). This sequence of events underlies C-type inactivation gating in KcsA and likely in other Kv channels (O/O to O/I transition). A network of water molecules that resides in a cavity behind KcsA's selectivity filter stabilizes its collapsed conformation; diffusion of water molecules into the inactivation cavity determines the rate and magnitude of C-type inactivation. Once the stimulus is terminated, the activation gate closes (O/I to C/I transition) although the selectivity filter remains collapsed for a short period of time, likely defined by the exit rate of water molecules from the inactivation cavity (*Ostmeyer et al., 2013*). Finally, the selectivity filter resets to the conductive conformation (C/I to C/O transition) completing the KcsA kinetic cycle (*Figure 8—figure supplement 1*.

Trapping KcsA in its open conformation has allowed the development of an atomic description of its O/O and O/I kinetic states. These two new kinetic intermediate snapshots complement, at high resolution, the existing structures for the C/O (high $K^+$-structure) and the C/I (low $K^+$-structure) conformations (*Zhou et al., 2001a*) and highlight a network of water molecules behind the channel's selectivity filter that stabilizes its collapsed conformation (*Figure 6a and b*). The structural description of this gating cycle stresses diffusion of water molecules into the inactivation cavity as a key determinant of the rate and magnitude of KcsA's C-type inactivation.

Finally, a recent Cryo-EM structural study of the human Ether-à-go-go-Related $K^+$ channel (hERG) suggested that the structural changes at the selectivity filter associated with C-type inactivation are subtle and might be different from those reported for KcsA (*Wang and MacKinnon, 2017*). Given hERG distinctive inactivation process, it is not surprising that the structure-function correlation underlying hERG inactivation is unique and different from the one displayed by KcsA and other voltage-gated $K^+$ channels (*Cuello et al., 2010a*).

## Materials and methods

### KcsA expression, purification and crystallization

KcsA (The Locked-open and closed conformations) cloned in the pQE70 expression vector were transformed in freshly-made XL10-gold competent cells by heat shock method. Cell cultures harvesting the plasmid to express the desired KcsA mutant were grown overnight at 37°C in the presence of 2% glucose and 0.4 mg/ml ampicillin. Next day, the overnight culture was diluted 100 times in 1 liter of LB media supplemented with 0.4 mg/ml and 0.2% glucose and grown at 37°C. Once the cells have reached an O.D. of 0.6, the cultures were cooled down to for 2 hr. Next, 10 mM $BaCl_2$, 0.4 mg/ml ampicillin and 0.4 mM IPTG were added to induce protein production at 29°C for ~18 hr (*Cuello et al., 2010b*). Next day, cells expressing KcsA were collected by centrifugation and resuspended in a buffer of the following composition: 50 mM Tris-Cl, pH 7.5, 150 mM KCl, 170 ug/ml

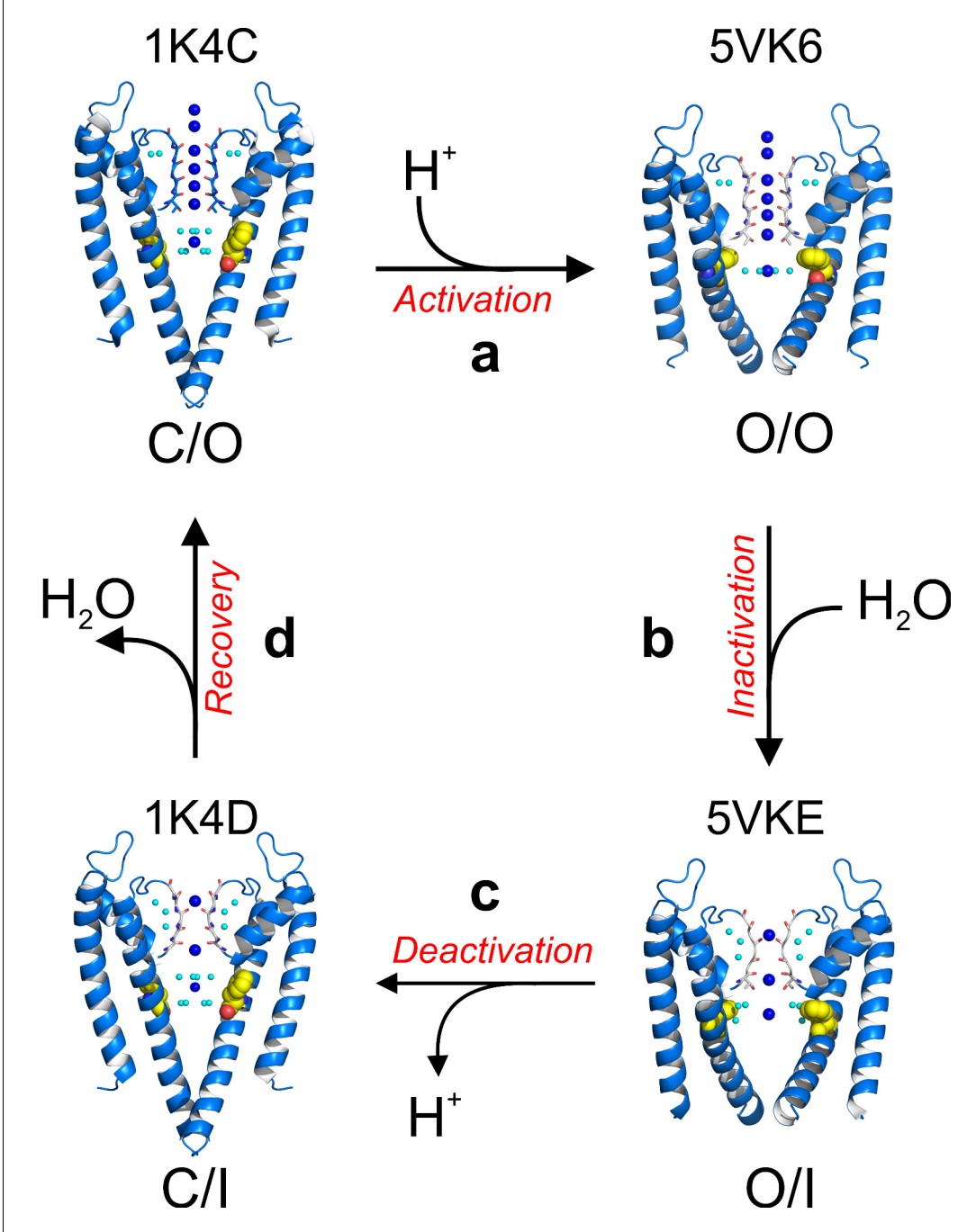

**Figure 8.** A structure-driven kinetic cycle for KcsA gating. The simplest KcsA kinetic cycle contains at least four distinct states, closed and prime-to-conduct (C/O), open and conductive (O/O), open and C-type inactivated (O/I) and closed and C-type inactivated (C/I). KcsA resting state (C/O) is best represented by the structure of its closed state in high-potassium concentration (PDB 1K4C). (a) Upon proton activation, the channel undergoes conformational changes at its activation gate, increasing its diameter to 22 Å, which allows K$^+$ ions to flow down their electrochemical gradient for about 200 ms, just before the onset of C-type inactivation. The Locked-open KcsA E71A mutant (PDB 5VK6) likely represents this short-lived open and conductive state (O/O). (b) A strong allosteric coupling between KcsA's activation gate and its filter determines the duration of its O/O state. Opening of the channel activation gate is communicated to the filter by mechanical deformation that propagates along a network of interacting amino-acid residues along the second membrane-spanning segment. Among many, Phe 103 undergoes conformational changes (yellow spheres) that make it clash with Thr 75 on the channel's pore helix, triggering a series of structural rearrangement at the channel's filter that yield a pore with two consecutive ion

*Figure 8 continued on next page*

*Figure 8 continued*
vacancies and collapsed or deep C-type inactivated. The Locked-open KcsA Y82A mutant structure (PDB 5VKE) recapitulates the most important features of KcsA's O/I state. (c) Decreasing intracellular proton concentration triggers KcsA deactivation gating, which involves structural rearrangements that close the channel's activation gate. For a brief period of time, KcsA exists with a collapsed filter and its activation gate closed. The structural model that best describes KcsA's C/I state is the closed structure in low $K^+$ concentration (PDB 1K4D). (d) Once the KcsA activation gate is closed, the filter remains collapsed for an undetermined period of time, though computational calculations have suggested that the exit of the 'inactivating waters' from the 'inactivation cavity' determines the recovery rate of the channel's filter.

DOI: https://doi.org/10.7554/eLife.28032.011

The following figure supplement is available for figure 8:

**Figure supplement 1.** A structural explanation of C-type inactivation gating at the KcsA's selectivity filter.

DOI: https://doi.org/10.7554/eLife.28032.012

phenylmethylsulfonyl fluoride (Buffer A). The cell suspension was disrupted by passing them three times through an ice-cold Emulsiflex C3. The broken cell suspension was spun down at 100,000 g for 1 hr and after discarding the supernatant, the cell membrane pellet was resuspended in buffer A and stored and −80°C.

A KcsA containing membrane preparation was solubilized with 20 mM dodecyl-maltoside for 2 hr, the insoluble material was spun down at 100,000 g and the supernatant was loaded into a pre-packed cobalt resin column. Next, the column was washed with 10 column volumes (CV) of buffer A, 10 mM imidazole and 1 mM DDM. Finally, KcsA was eluted with buffer A, 1 mM DDM and 400 mM imidazole. Chymotrypsin-cut KcsA was complexed with an antibody fragment used for crystallization purposes and purified in 5 mM DM. Crystal trials were set up by the sitting-drop method in 24–27% PEG400 (v/v), 50 mM magnesium acetate, 50 mM sodium acetate (pH 5.4–6.0) at 20°C (*Cuello et al., 2010b*).

## Patch-clamp studies

Locked-open KcsA and the closed channel containing single point mutations that enhance or eliminate C-type inactivation gating were reconstituted in Asolectin liposomes. In brief, channels were reconstituted at a 1 to 100 for macroscopic currents or 1 to 5000 protein-to-lipid ratio for single-channel recordings, in Asolectin liposomes made in the following buffer: 200 mM KCl and 5 mM MOPS-buffer at pH 7.0 (buffer B) and incubated overnight with bio-beads from Bio-Rad. Next day, KcsA containing proteoliposomes were harvested by ultracentrifugation at 100,000 g for 1 hr. Then, the pellet was resuspended in 60 μL of buffer B and three drops of this proteoliposomes suspension were placed on a microscope glass slides and dehydrated overnight inside a desiccation chamber. Next day, the dried drops were rehydrated with 20 μl at 4°C overnight. Normally, the samples were rehydrated for ~24 hr yielding giant liposomes appropriate for patch-clamp experiments. KcsA channel activity was recorded in symmetrical 5 mM MOPS at pH 4.0 in the presence of 200 mM KCl or RbCl. KcsA currents were recorded with a patch-clamp amplifier Axopatch 200 B, and currents were sampled at 40 kHz with an analogue filter set to 10 kHz. Patch pipettes, after fire polishing, displayed a resistance of 2.0 MΩ (they were filled with 200 mM KCl and 5 mM MOPS-buffer at pH 4.0). Data Analysis: to calculate the Tau for inactivation a single exponential curve was fitted to the data, and the reported values are the mean ±SEM of 5 to 10 independent experiments. To estimate the nominal open probability of the Locked-open E71A and Y82A mutants, steady-state single-channel recordings at pH 8 or 3 were measured. The open probability was calculated from Gaussian fits to the peaks of all point histograms of the single-channel recording at each measured pH.

## Crystallographic analysis

Crystallographic data were collected at the beamlines 23ID (GMCA) and Beamline 14-ID-B at the Advance Photon Source (Argonne) and at the beamline 14–1 at the Stanford Synchrotron Radiation Laboratory (SSRL) and processed with HKL2000 (*Otwinowski and Minor, 1997*). The phase of all KcsA structures was obtained by molecular replacement using as a search model, the antibody fragment bound to the KcsA-OM (PDB = 35FW). A structural model for the locked-open KcsA was built from scratch using Coot (*Emsley and Cowtan, 2004*) for modeling followed by iterative cycles of

refinement using Phenix (*Adams et al., 2010*). For the B-factors analysis a new structure for KcsA in the closed state was solved in house at the same resolution and using the same parameters during the refinement process of the Locked-open E71A mutant. The figures on this paper were made using Pymol (https://pymol.org) and the statistical analysis was done in Origin (OriginLab).

## Acknowledgements

We thank Luis Reuss for insightful discussions and comments during the data analysis aspect of this project. We thank Olivier Dalmas for helping and assisting during the initial data collection of this project. We thank Vukica Srajer at Beamline 14-ID-B (Advanced Photon Source) and Silvia Russi at the Stanford Synchrotron Radiation Laboratory Beamline (SSRL) 14–1 for assistance at the synchrotron. This work was supported by NIH U54 GM087519 to EP and NIH 1RO1GM097159-01A1 and Welch Foundation [BI-1757] to LC.

## Additional information

### Funding

| Funder | Grant reference number | Author |
| --- | --- | --- |
| National Institute of General Medical Sciences | U54 GM087519 | Eduardo Perozo |
| Welch Foundation | BI-1757 | Luis G Cuello |
| National Institute of General Medical Sciences | 1RO1GM097159-01A1 | Luis G Cuello |

The funders had no role in study design, data collection and interpretation, or the decision to submit the work for publication.

### Author contributions

Luis G Cuello, Conceptualization, Formal analysis, Supervision, Funding acquisition, Validation, Investigation, Writing—original draft, Project administration, Writing—review and editing; D Marien Cortes, Investigation, Methodology; Eduardo Perozo, Conceptualization, Resources, Data curation, Formal analysis, Supervision, Funding acquisition, Investigation, Methodology, Writing—original draft, Project administration, Writing—review and editing

### Author ORCIDs

Luis G Cuello (iD) http://orcid.org/0000-0001-5192-7568
Eduardo Perozo (iD) http://orcid.org/0000-0001-7132-2793

### Decision letter and Author response

Decision letter https://doi.org/10.7554/eLife.28032.015
Author response https://doi.org/10.7554/eLife.28032.016

## Additional files

### Supplementary files

• Transparent reporting form
DOI: https://doi.org/10.7554/eLife.28032.013

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
