## [Decision Letter]

Thank you for submitting your article "The gating cycle of a K^+^ channel at atomic resolution" for consideration by *eLife*. Your article has been reviewed by three peer reviewers, one of whom, Kenton J Swartz (Reviewer #2), is a member of our Board of Reviewing Editors, and the evaluation has been overseen by Richard Aldrich as the Senior Editor. The following individual involved in review of your submission has agreed to reveal their identity: Roderick MacKinnon (Reviewer #1).

The reviewers have discussed the reviews with one another and the Reviewing Editor has drafted this decision to help you prepare a revised submission.

Summary:

This manuscript addresses conformational states that underlie gating in the KcsA K^+^ channel. This channel opens transiently following steps in pH from 8 to 3 and then inactivates (some would say desensitizes) partly. Activation is associated with opening of an inner helix gate and inactivation as shown here seems associated with conformational changes at the selectivity filter. This conclusion is supported by demonstration that KcsA with a trapped open inner helix gate (using disulfide bridges) still gates. Mutations near the selectivity filter that increase or decrease inactivation favor different conformations in the setting of the trapped open inner helix gate. The same lab determined similar structures at an earlier date but these are at higher resolution. The higher resolution was achieved through the disulfide bridges, which limited heterogeneity. This study makes a compelling case for two distinct gates in KcsA that are both important for gating in KcsA. The following are requested revisions for improving the manuscript.

Essential revisions:

1) Unless we are mistaken, the functional characterization of the Y82A and E71A mutants was not done in the background of the disulfide bonded OM. This is not trivial to do because presumably these constructs would not be responsive to low pH because the lower pH controlled gate is locked open. However, it would be nice to know that they are not responsive to low pH and then to see single channel recording similar to those in Figure 2 that are provided for the OM alone. It would be nice to see that the E71A-OM has a high Po and that the Y82A-OM has a low Po, etc. Also, the functional data that are provided for E71A and Y82A in Figure 3 and the OM in Figure 2 are all single observations and the authors should provide population data.

2) In subsection “Trapping KcsA with its activation gate in the open conformation” of the manuscript, the authors refer to thermal denaturation experiments in Figure 2, but no wild-type KcsA control is presented in the ms nor is a result for WT KcsA cited from the literature; this should be addressed. It's probably also worth noting in the text that the 28-118 disulfide bond is formed between residues in adjacent KcsA subunits. The authors should therefore note whether the sample buffer in Figure 2 contained DTT and/or β-mercaptoethanol, or if reducing agents were present during heating of the sample (no experimental details are provided).

3) We are skeptical about the discussion of B factors (subsection “Atomic resolution structure of KcsA open and conductive” paragraph three). How did the B factors compare in the 2 structures outside the selectivity filter? More importantly, one structure has a mutation in a residue surrounding the filter (E71A) and the other does not. Because such a mutation could affect the filter, we would hesitate to give too much interpretation to their B factor differences. Moreover, it's probably not appropriate to directly compare B-factors from at two different resolutions (2.0 Å for the 1K4C structure, 2.25 Å for the 5VK6 structure), in which the structures may have been refined using different parameterization. In addition, the B-factors plotted in Figure 5 appear to include those of both main chain and side chain atoms. Because B-factors for side chain atoms may be less well-defined and more subject to bias than main chain atoms, it would likely be more useful to limit the comparison to main chain atoms only. To further minimize bias in the comparisons, both data sets would need to be truncated to the same resolution (and probably same redundancy) and then re-refined, and then the authors will need to provide a strong rationale for the statistical distribution of the B-factors and provide some error estimates of these values to justify their argument. In some previous work, investigators have justified using a normalization of the B-factors, so this may be a valid approach.

4) The authors state that water exchange is a key determinant in the rate and magnitude of inactivation. This could be true but the evidence is not very strong. Does changing the activity of bulk water alter inactivation, as one would predict on the basis of this assertion? The authors should provide additional experimental support for this conclusion.

5) Subsection “Atomic resolution structure of KcsA open and deep C-type inactivated” and further in the Discussion: historically, the name C-type inactivation was given to the process of inactivation in Shaker K channels following removal of the N-terminal inactivation peptide. How do we know that the process of inactivation in KcsA (or desensitization following opening after a pH step) is the same as inactivation following voltage dependent gate opening in Shaker? We think it would be fine to motivate this work with previous studies on C-type inactivation in Shaker or other K channels, but at the end of the day this body of work is about KcsA and the authors should restrict their implications on mechanism to that protein and be more circumspect throughout.

6) Related to point 3, the discussion contained a large section opposing another study of inactivation based on D-alanine mutations that would seem to prevent the inactivated conformational change observed here. The authors state in the Discussion: "Indeed, recent computational analyses of a D-Alanine "rigidized" selectivity filter demonstrated that the selectivity filter is able to relax to a partially constricted conformation, with the D-Ala side chains at position 77 occluding the permeation pathway (Li et al., 2017). Therefore, the 77D-Ala substitution does not lead to a rigid selectivity filter, and crucially, does not preclude the transition to a constricted conformation for the selectivity filter." This argument supposes the correctness or accuracy of the computational studies. Perhaps more importantly, don't you think it is possible that inactivation in the selectivity filter, or its inability to conduct, might be associated with a number of different filter conformations (i.e. not all inactivated filters/states look the same)?

7) What is the distinction between a C-type inactivated state and a deep C-type inactivated state? Please explain the difference.

8) Subsection “Atomic resolution structure of KcsA open and deep C-type inactivated”: "Crystals of the O/I-state diffracted ~ 2.4 A with good refinement statistics, R_work_ = 0.1948 and R_free_ = 0.2267 and its structure was solved by molecular replacement…" Crystals do not diffract with good or bad refinement statistics. You mean the model exhibited those residuals when compared to the data. In addition, the crystallographic data table (Table 1) is incomplete. The mean R_merge_ values for each data set are conspicuously high. Many investigators have switched to R_pim_ or CC1/2 as an indicator of overall data quality, but these are not provided. In addition, the value of i/sigma appears to only correspond to the high-resolution shell, and the overall i/sigma is not provided. Finally, the redundancy appears to only correspond to overall redundancy, and not high-resolution shell. Without a completed data table, it is difficult for the reader to independently evaluate the X-ray data quality or make comparisons among the data sets analyzed in this manuscript. For the refinement statistics, the authors list the "Wilson B-factor", which is estimated from the experimental data. The authors should instead provide the mean b-factors for atoms in each model; these are usually provided separately for the protein, water, and ligand atoms.

9) In the Discussion, the authors summarize their findings as if they have completed a mathematical proof: "Taking together existing data with the present high resolution structures we can recapitulate the process of C-type inactivation in the framework of a mechanistic model (Figure 8 and Figure 8—figure supplement 1)". Would it not be better to say, “on the basis of data presented we propose the following…” After all, we all know how tricky it can be to figure out what is really happening in these molecules!

---

## [Author Response]

Essential revisions:1) Unless we are mistaken, the functional characterization of the Y82A and E71A mutants was not done in the background of the disulfide bonded OM. This is not trivial to do because presumably these constructs would not be responsive to low pH because the lower pH controlled gate is locked open. However, it would be nice to know that they are not responsive to low pH and then to see single channel recording similar to those in Figure 2 that are provided for the OM alone. It would be nice to see that the E71A-OM has a high Po and that the Y82A-OM has a low Po, etc. Also, the functional data that are provided for E71A and Y82A in Figure 3 and the OM in Figure 2 are all single observations and the authors should provide population data.

The reviewers are correct. Originally, we considered that sampling the behavior of the pH-activated channel would provide a close sampling of the gating transitions at the selectivity filter in either of the mutants. And while we still stand by that assessment, we do agree that formally, these electrophysiological measurements should be carried out on disulfide-cross-linked OM backgrounds (even if it is more difficult). Single channel traces are now shown in Figure 3, with Figure 3 depicting overall Open Probability behavior for the population active in the patch (NPo). As expected, when the inner bundle gate is crosslinked, single channel activity becomes pH independent, and solely defined by the mutation at the selectivity filter/pore helix.

2) In subsection “Trapping KcsA with its activation gate in the open conformation” of the manuscript, the authors refer to thermal denaturation experiments in Figure 2, but no wild-type KcsA control is presented in the ms nor is a result for WT KcsA cited from the literature; this should be addressed. It's probably also worth noting in the text that the 28-118 disulfide bond is formed between residues in adjacent KcsA subunits. The authors should therefore note whether the sample buffer in Figure 2 contained DTT and/or β-mercaptoethanol, or if reducing agents were present during heating of the sample (no experimental details are provided).

Thermal denaturation assays have been performed in our lab to assess overall stability of KcsA mutants for many years (first described in Cortes & Perozo, Biochemistry 36(33):10343-52 (1997). We followed the same protocol here. We now include the control experiment in Figure 2, state that disulfide bonds are indeed *inter*-subunit (legend, Figure 3), and include the methodological details of the experiment (Materials and methods section).

3) We are skeptical about the discussion of B factors (subsection “Atomic resolution structure of KcsA open and conductive” paragraph three). How did the B factors compare in the 2 structures outside the selectivity filter? More importantly, one structure has a mutation in a residue surrounding the filter (E71A) and the other does not. Because such a mutation could affect the filter, we would hesitate to give too much interpretation to their B factor differences. Moreover, it's probably not appropriate to directly compare B-factors from at two different resolutions (2.0 Å for the 1K4C structure, 2.25 Å for the 5VK6 structure), in which the structures may have been refined using different parameterization. In addition, the B-factors plotted in Figure 5 appear to include those of both main chain and side chain atoms. Because B-factors for side chain atoms may be less well-defined and more subject to bias than main chain atoms, it would likely be more useful to limit the comparison to main chain atoms only. To further minimize bias in the comparisons, both data sets would need to be truncated to the same resolution (and probably same redundancy) and then re-refined, and then the authors will need to provide a strong rationale for the statistical distribution of the B-factors and provide some error estimates of these values to justify their argument. In some previous work, investigators have justified using a normalization of the B-factors, so this may be a valid approach.

This is an important point brought by the reviewers. Formally, an approach to compare Debye-Weller B temperature factors in structures of different resolutions (and hence B factors) has been to artificially increase the overall values of the highest resolution structure. However, these comparisons are always limited by the worst-resolution data of the pair. Alternative, one could subtract the average B factor for a common stretch of residues in each case and compare, not the actual B factor values, but its variance (see Cordero-Morales et al., 2006). We have indeed followed the reviewer’s suggestions and carried out a de novo solution of wild-type KcsA, refined in house with the same parameters used during the refinement of the Locked-open E71A, at equivalent resolutions. This dataset was then used to redo the B-Factor analysis, but now with the normalized B-factors of the α carbons of amino acids of the selectivity filter. This newly generated comparison, between closed and open KcsA is shown in a new Figure 5.

4) The authors state that water exchange is a key determinant in the rate and magnitude of inactivation. This could be true but the evidence is not very strong. Does changing the activity of bulk water alter inactivation, as one would predict on the basis of this assertion? The authors should provide additional experimental support for this conclusion.

We have already carried out an earlier set of experiments to test this issue experimentally (in addition to extensive molecular dynamics simulations) see Ostmeyer et al., 2013. In this publication, we showed that reducing the occupancy of the buried water molecules at the filter (by imposing a high osmotic stress) accelerates the rate of recovery, as measured from the recovery rate of inactivation in the presence of a high concentration osmolyte solution (2M sucrose). The experiment provides strong, and direct experimental evidence supporting the hypothesis that water access to cavities behind the selectivity filter is a key driver of the kinetics on inactivation and inactivation repriming in KcsA. See Figure 4 from Ostmeyer et al.

5) Subsection “Atomic resolution structure of KcsA open and deep C-type inactivated” and further in the Discussion: historically, the name C-type inactivation was given to the process of inactivation in Shaker K channels following removal of the N-terminal inactivation peptide. How do we know that the process of inactivation in KcsA (or desensitization following opening after a pH step) is the same as inactivation following voltage dependent gate opening in Shaker? We think it would be fine to motivate this work with previous studies on C-type inactivation in Shaker or other K channels, but at the end of the day this body of work is about KcsA and the authors should restrict their implications on mechanism to that protein and be more circumspect throughout.

We have to disagree with this assessment. First of all, others and we have demonstrated that the time-dependent reduction in open probability observed in KcsA is not due to a “desensitization” process, which would somehow affect the pH-dependence of the channel but to the existence of a second gate at KcsA’s selectivity filter (an inactivation gate). This is not a controversial point (see Cordero-Morales et al., 2006).

However, regarding nomenclature, perhaps this is an issue of semantics: while it is indeed correct that C-type inactivation was detected (and defined) in voltage dependent K^+^ channels as the time dependent reduction in ion current still present in the absence of fast N-type inactivation, there is ample evidence over the past 20+ years that point to the functional equivalence between the KcsA inactivation gating and that seen in voltage dependent K^+^ channels. The details differ, of course, but there are many common observations that strongly suggest, at the very least, a “kinship” between these evolutionarily related channels:

– Inactivation in KcsA displays the same kinetic relations (transition into and out of C-type inactivated state under a two-pulse protocol);

– Same external dependence of inactivation kinetics on external K^+^ concentrations;

– Same behavior with conductive and non-conductive ions: Rb^+^, plus Tl^+^, NH_4_^+^ Cs^+^ (unpublished);

– Equivalent mutations that enhance C-type inactivation in eukaryotes do lock in the pinched filter conformation in KcsA;

– Mutations that inhibit C-type inactivation in eukaryotes lead to conductive filter conformations in KcsA.

We argue that the mechanism of inactivation in KcsA is at least related to those present in more complex eukaryotic channels. We all look for commonalities among evolutionarily related systems, but the jury is definitely out on whether there will be a “standard” C-type inactivation mechanism or different “classes” of it, depending on the pore domain. Our work points to the initiation of inactivation in KcsA as the loss of a potassium ion at the second binding site, accompanied by changes in pore diameter neat Glycine 77. This would lead to increase in the energy barrier for K^+^ translocation between biding sites. Still, there might be other ways to add a (relatively) subtle energy barrier to inhibit K^+^ flow, so that in some instances the Kv channel’s filter might not need to adopt the same pinched conformation to inactivate (as might be the case in the I1 conformation of KcsA). Ultimately, the observation of a similarly pinched filter in a eukaryotic pore domain (something that so far has not been observed) might help settle the score toward a “common mechanism”. Consequently, we have “toned down” the direct equivalence between eukaryotic C-type inactivation and KcsA inactivation and addressed the present results and interpretation within the context of the KcsA results and mechanism.

6) Related to point 3, the discussion contained a large section opposing another study of inactivation based on D-alanine mutations that would seem to prevent the inactivated conformational change observed here. The authors state in the Discussion: "Indeed, recent computational analyses of a D-Alanine "rigidized" selectivity filter demonstrated that the selectivity filter is able to relax to a partially constricted conformation, with the D-Ala side chains at position 77 occluding the permeation pathway (Li et al., 2017). Therefore, the 77D-Ala substitution does not lead to a rigid selectivity filter, and crucially, does not preclude the transition to a constricted conformation for the selectivity filter." This argument supposes the correctness or accuracy of the computational studies. Perhaps more importantly, don't you think it is possible that inactivation in the selectivity filter, or its inability to conduct, might be associated with a number of different filter conformations (i.e. not all inactivated filters/states look the same)?

The argument is clearly supported by the computational calculations, but it is not necessarily dependent on their accuracy. We make the case that simply, the complexity of behavior of the unnatural substitutions at the selectivity filter makes it impossible to univocally conclude that “the constricted conformation of the selectivity filter is not the C-type inactivated state” even when some reduction in open probability is observed in the presumably rigid D-Ala mutation.

We buttress our argument, not only with solid computational work (allowing molecular explorations that are very difficult to evaluate experimentally), but also from recent 2D infrared experiments done by us (and in collaboration with, among others, F Valiyaveetil). In this work, we provide extensive evidence for the complexity of conformations (as coupled vibrational modes) at the selectivity filter and its dependence on the state of the intracellular gate (see, Kratochvil HT et al. JACS 139(26):8837-8845 (2017)). Our findings indicate that changes in the intracellular gate are linked to changes in the selectivity filter conformational heterogeneity that are distinct to those triggered by K^+^ depletion. The existence of previously undetected conformational subensembles appear to underlie changes in ion affinity at the selectivity filter and suggest that this complexity does not disappear (just changes) in the presence of unnatural amino acids. More recent computational work (Li et al., now in press) provides strong evidence for an alternative pinched state (nonconductive) in the D-Ala KcsA mutant. We have reworked this section to provide a more inclusive set of interpretations to the inactivation process when D-Ala is part of the selectivity filter.

7) What is the distinction between a C-type inactivated state and a deep C-type inactivated state? Please explain the difference.

We defined the deep C-type inactivated state as that reached after steady state once the inner bundle gate opens (t →∞). With this definition, we can have a better equivalent to the determined crystal structures, while making a distinction with other (presumably non-conductive) states observed when the inner bundle gate is not fully open (see Cuello et al., 2010).

8) Subsection “Atomic resolution structure of KcsA open and deep C-type inactivated”: "Crystals of the O/I-state diffracted ~ 2.4 A with good refinement statistics, R_work_ = 0.1948 and R_free_ = 0.2267 and its structure was solved by molecular replacement…" Crystals do not diffract with good or bad refinement statistics. You mean the model exhibited those residuals when compared to the data. In addition, the crystallographic data table (Table 1) is incomplete. The mean R_merge_ values for each data set are conspicuously high. Many investigators have switched to R_pim_ or CC1/2 as an indicator of overall data quality, but these are not provided. In addition, the value of i/sigma appears to only correspond to the high-resolution shell, and the overall i/sigma is not provided. Finally, the redundancy appears to only correspond to overall redundancy, and not high-resolution shell. Without a completed data table, it is difficult for the reader to independently evaluate the X-ray data quality or make comparisons among the data sets analyzed in this manuscript. For the refinement statistics, the authors list the "Wilson B-factor", which is estimated from the experimental data. The authors should instead provide the mean b-factors for atoms in each model; these are usually provided separately for the protein, water, and ligand atoms.

Agreed. The crystallographic tables are now fully populated and reformatted according to the reviewer’s suggestions.

9) In the Discussion, the authors summarize their findings as if they have completed a mathematical proof: "Taking together existing data with the present high resolution structures we can recapitulate the process of C-type inactivation in the framework of a mechanistic model (Figure 8 and Figure 8—figure supplement 1)". Would it not be better to say, “on the basis of data presented we propose the following…” After all, we all know how tricky it can be to figure out what is really happening in these molecules!

Fair enough! We have changed the text to be less deterministic and allow for the complexities of a still not fully understood process. Our explanations have been constrained to implications and interpretation relevant to KcsA inactivation while considering the possibility that our findings might apply to other Kv channels.